# Occult hepatitis B virus infections and risk factors among school-going adolescent voluntary blood donors in Kwale County Kenya, January 2020–June 2021: Cross sectional study

**Peter Kitemi Wahome[1,2], Polly Kiende[3], Rocky Jumapili Nakazea[2], Narcis Mwakidedela Mwasowa[2], Gibson Waweru Nyamu[2]***

1 Technical University of Mombasa, Mombasa, Kenya, 2 Department of Health, Kwale County, Kenya, 3 Department of Health, Tharaka Nithi, Chuka, Kenya

* wawerugibson2015@gmail.com

**Data Availability Statement:** All relevant data are within the paper and its Supporting Information files.

## Abstract

### Background

Occult hepatitis B virus (HBV) infections remain a safety concern worldwide. The prevalence in Kenya ranges from 2.6% to 4.4% among secondary school-going voluntary blood donors. This study estimated the prevalence of occult HBV infections among school-going voluntary blood donors through donations made to Kwale Satellite Blood Transfusion Center (KSBTC).

### Methods

This was a retrospective cross-sectional study on data collected by the KSBTC between January 2020 and June 2021 among secondary school-going voluntary blood donors. Data were collected in MS Excel 2013 and analyzed in Epi Info 7. Descriptive statistics were calculated and we compared donors with positive Hepatitis B surface antigen (HBsAg) to those with negative HBsAg. Crude Prevalence Odds Ratios (cPOR) at 95% confidence intervals (CI) were calculated to identify factors associated with positive HBsAg.

### Results

A total of 613 records were analyzed. The mean age of the donors was 19.1 years (± 1.8 years), there were 457 males (74.5%), 502 individuals were in the age group 18–25 years (82.3%), and the mean hemoglobin level was 14.1 g/dl (±1.6 g/dl). First-time blood donors made up 84.8% of all donors (513/605) and the mean inter-donation period was 20 months (±5.8 months) for repeat donors. The sero-positivity for HBsAg was 8.8% (54/613). Age category 16–17 years with positive HBsAg were 10.2% (11/108), femaleswere10.9% (17/156), and first-time donors were 9.4% (48/513). On bivariate analyses, first-time blood donors were 1.5 times more likely to test positive for HBsAg compared to repeat donors (cPOR = 1.5, 95% CI 0.61–3.57). Females were 1.4 times more likely to test positive for HBsAg compared to male donors (cPOR = 1.4, 95% CI 0.76–2.54).

**Funding:** The authors received no specific funding for this work.

**Competing interests:** The authors have declared that no competing interests exist.

**Abbreviations:** CI, Confidence intervals; cPORs, Crude odds ratios; HBsAg, Hepatitis B surface antigen; OBI, Occult Hepatitis Infections; IDU, Intravenous Drug Use; KNBTS, Kenya National Blood Transfusion Service; KSBTC, Kwale Satellite Blood Transfusion Center; MCRH, Msambweni County Referral Hospital; TTIs, Transfusion transmissible infections; WHO, World Health Organization.

## Conclusions

The majority of the voluntary blood donors were males and the majority of occult HBV infections came in the first-time blood donor group. We recommend increasing targeted recruitment of repeat donors by encouraging healthy first-timer donors to be regular donors, and suggest this population should be vaccinated against HBV infections.

## Background

Blood and blood products are essential commodities used to save lives. The safety of blood and blood products is of utmost importance as patients are often transfused in a vulnerable health state. Hepatitis B virus (HBV) being a transfusion-transmissible infection, it remain hazardous in donated blood in spite of the current blood donation safety developments put in place [1]. It is also compounded by the fact that donor blood is majorly sourced from adolescents mainly in learning institutions [2]. In this period in life the adolescence are highly vulnerable to the disease as this is the period of the beginning of very active and risky sexual behaviors and other practices that increase the risks of exposure to HBV infection [3]. A person with occult hepatitis B infection (OBI) has a risk of reactivation during immunosuppression and due to lack of detectable HBsAg there is an ongoing risk of transmission to others [4]. Globally, OBI is ≥8% prevalence in endemic regions and is high in populations at high risk of infection including the HIV infected or people who inject drugs [5]. A majority of people are unaware of their HBV infection, and often present with advanced disease [13], as noted in Nigeria where 5.4% of the donors had anti-HBcIg M as the only serological evidence of HBV infection [14]. In Africa, the prevalence of OBI is estimated to be between 7–50% among blood donors or healthcare workers [6, 7], and 6–30% in HIV infected populations [8]. High rates of OBI was reported in HIV-positive individuals initiating antiretroviral therapy in Botswana [9]. In sub-Saharan Africa, 12.5% of patients who receive blood transfusion are at risk of post-transfusion OBI [10].

Studies conducted in Kenya related to sero-prevalence of HBV have focused on HIV patients, blood donors in general, and the general population with a few focusing on adolescents in Kenya. One study in Western Kenya showed a sero-prevalence of 3.4% HBsAg among school-going adolescent donors aged 18 to 25 years [11]. In Kwale County, while it is known that the annual demand for blood is approximately 9000 units according to the 2019 census population [12], no such study has been conducted in this area and therefore the prevalence of OBI is not known. This study aims to estimate the magnitude of OBI and the possible risk factors among school-going adolescent donors in Kwale County. The results will help programs undertaken by the Kwale Satellite Blood Transfusion Centre (KSBTC) and non-government agencies to inform policy decisions and planning with an aim of supporting existing structures and improving blood safety and availability.

## Methods

### Study setting

This study was conducted among voluntary school-going adolescent donors through mobile site collection in the various secondary schools in Kwale County. The KSBTC is located at Msambweni County Referral Hospital (MCRH) (Fig 1).

Trained medical personnel made appointments with leaders of schools by writing letters, emailing, and making phone calls to explain the purpose and the intention of conducting the

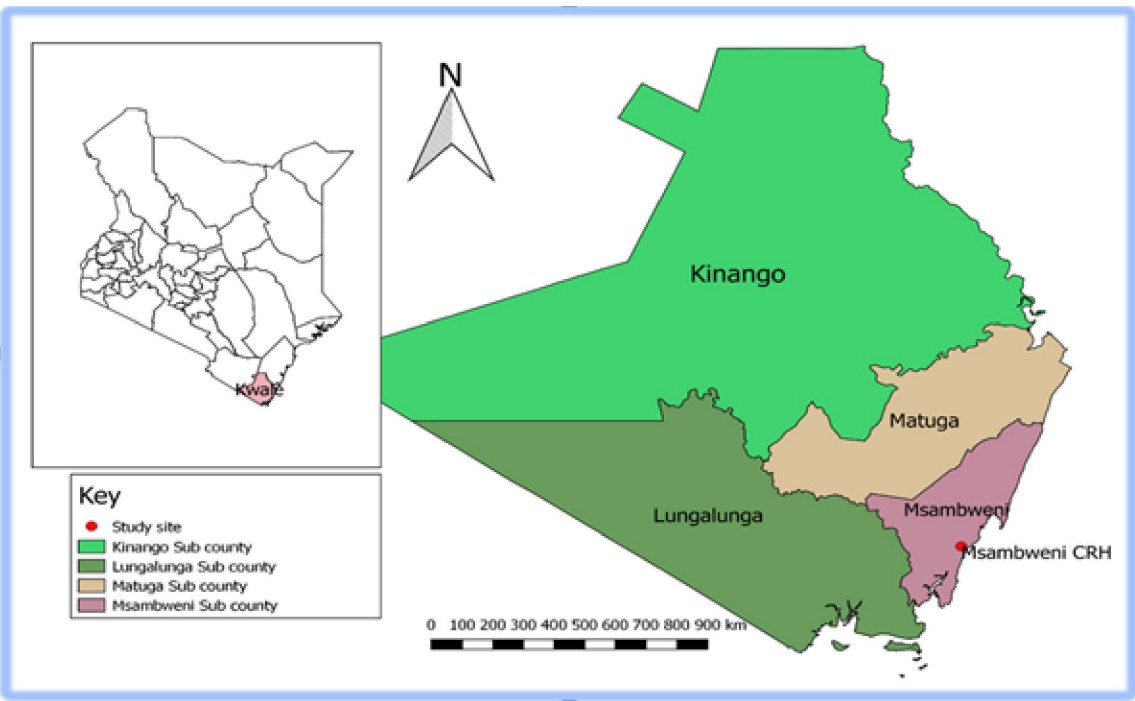

**Fig 1. Map showing the study site—Kenyan map.** Counties sampled (Map developed by Author using QGIS Version 2.18.10) with geographical data was obtained from https://africaopendata.org/dataset/kenya-counties-shapefile and data on geo—coordinates and category of health facility (Msambweni County Referral Hospital) was obtained from Kenya Master Health facility list http://kmhfl. health.go.ke/.

blood drive donation in their respective secondary schools. Once permission was granted, the pre-donation health talk was given to the prospective donors. This covered the importance of blood donation, the characteristics of blood donors, and donor deferrals. The prospective donors were subjected to the national blood transfusion questionnaires for risk assessment and those who met the criteria for blood donation as per the national guidelines were allowed to donate and signed a consent form after having been informed about testing of several TTIs namely, HBV, HIV, HCV, and syphilis. The collected blood was then packed in a cooler box and shipped to the KSBTC for screening of TTIs. Blood samples collected from each subject were placed into sterile serum collection bottles and centrifuged at 1,500 revolutions per minute for 15 minutes. The separated serum was placed in sterile cryovials and stored at 2–8˚C. Thereafter, all blood tests were done at the KSBTC laboratory using Murex HBsAg version 3 (Manufactured by Diasorin S.p.A UK Branch). All the tests, which showed optical density values above the cut-off value as calculated by the manufacturer's instructions, were considered reactive for HBsAg. The quality controls were conducted for every run where three control wells for both positive and negative test results were included, known positive and negative samples were also included and the known controls/samples were treated and ran simultaneously with donor's samples.

## Study design and population

This study was a retrospective, cross-sectional, secondary analysis of the archived donor's clinical forms and data from the KSBTC between January 2020 and June 2021 to estimate the prevalence of HBsAg among voluntary school-going adolescent blood donors. The study included

voluntary school-going adolescent blood donors who met the KNBTS donation requirements [13].

- The donor had to be aged between 16 and 65 years for donation though for inclusion in the study one had to be aged between 16–25 years.

- The donor had to have a body weight of 50 kg or more.

- The individual's hemoglobin of not less than 12.5 g/dl.

- Informed written consent to participate in the study.

The study excluded students at secondary schools in the study period that donated blood for family/friends at the KSBTC and those who didn't meet the KNBTS donation criterion.

## Sample size determination

Cochran's formula [14] was used to calculate the sample size required to estimate the prevalence of HBsAg among voluntary school-going adolescent blood donors. The study assumed a 95% confidence interval, 80% power, prevalence of HBsAg 50%, and we adjusted by 10% to account for missing or incomplete data. We calculated the desired sample size as 424 participants; however, considering that period reports were evaluated in this study, a census-type sampling technique was used. A larger sample size (n = 613) was included and analyzed to allow for adequate statistical power.

## Data management and analysis

The data were entered into MS Excel 2013, cleaned, and double-checked before analysis using Epi-info 7 statistical package (CDC, Atlanta, USA). Frequencies and proportions were calculated for categorical variables. Means, standard deviations (SD) of continuous symmetrical variables, medians, and interquartile ranges for the continuous asymmetrical variables were calculated. We conducted a bivariable analysis and calculated the crude odds ratios (cPORs) and their 95% confidence intervals (CI) and considered p-values $\leq 0.05$ as statistically significant. Variables collected included age, sex, number of donations, religion, secondary schools, and HBsAg status as the outcome variable. We compared the HBsAg positive and negative participants.

## Ethics approval and consent to participate

Ethical clearance was obtained from the Technical University of Mombasa Ethics Review Committee (TUM ERC BSC/040/2020). Permission was granted to conduct the study by the Kwale County Department of Health (CG/KWL/6/5/1/CECM/Vol/1/15) and the director of KSBTC. Study participants informed consent was not required in this case, as this is secondary data obtained from an electronic database, and is impossible to track back blood donors for consent, however, the study participant's identity was fully anonymous during data acquisition and analysis. Also, this study included voluntary donors only and before blood donation is done every prospective donor has to consent to the testing of several TTIs namely; HBV, HIV, HCV, and syphilis. The source documents were stored in a lock and key cupboard and the electronic version was stored in password-protected computers.

## Results

### Voluntary blood donor's social-demographics

A total of 613 voluntary secondary school-going adolescent blood donors were included in the analysis. The mean age was 19.1 years (± 1.8 years), there were 457 males (74.5%), 502 donors

**Table 1. Distribution of blood group types among voluntary school-going adolescents' blood donors at KSBTC, January 2020-June 2021 (n = 605).**

| Blood groups types | Frequency | Percent |
|---|---|---|
| O Rhesus Positive | 343 | 56.7 |
| A Rhesus Positive | 125 | 20.7 |
| B Rhesus Positive | 107 | 17.7 |
| AB Rhesus Positive | 19 | 3.1 |
| O Rhesus Negative | 8 | 1.3 |
| A Rhesus Negative | 2 | 0.3 |
| B Rhesus Positive | 1 | 0.2 |

were aged between 18–25 years (82.3%), and Muslims comprised 335 individuals (54.9%). The mean Hb was 14.1 g/dl (±1.6 mg/dl). The most frequent donor blood groups were 56.7% O Rhesus positive (343/605), 20.7% A Rhesus positive (125/605) and 17.7% B Rhesus positive (107/605) (Table 1).

The number of first-time donors was 513 (84.8%) and for donors who were donating for the second time were 92 (15.2%). The mean month for the inter-donation period was 20 months (±5.8 months).

## Prevalence and factors associated with HBsAg positivity

The sero-positivity for HBsAg among the voluntary blood donors was 8.8% (54/613). The characteristics of donors with positive HBsAg were as follows: the age group 16–17 years comprised 10.2% (11/108), females comprised 10.9% (17/156), Christians made up 9.1% (25/275) and first-time donors were 9.4% (48/513) (Table 3). Regarding secondary school HBsAg positivity distribution, Lukore secondary had a positivity rate of 25.0% (8/32), Muhaka secondary had a positivity rate of 23.1% (3/13), and Waa Boys secondary had 10.2% (12/118) positivity rate (Table 2).

**Table 2. Distribution of HBsAg positivity among secondary schools at KSBTC, January 2020-June 2021 (n = 613).**

| Secondary Schools | N = 613 | Positive | Percent |
|---|---|---|---|
| Lukore Secondary | 32 | 8 | 25 |
| Muhaka Secondary | 13 | 3 | 23.1 |
| Magaoni Secondary | 32 | 6 | 18.8 |
| Waa Girls Secondary | 43 | 7 | 16.3 |
| Mwanambeyu Secondary | 15 | 2 | 13.3 |
| Kingwende Secondary | 23 | 3 | 13 |
| Dori Girls Secondary | 19 | 2 | 10.5 |
| Waa Boys Secondary | 118 | 12 | 10.2 |
| Kwale High Secondary | 91 | 7 | 7.7 |
| Madago Secondary | 16 | 1 | 6.3 |
| Gombato Secondary | 20 | 1 | 5 |
| Mwereni Secondary | 26 | 1 | 3.9 |
| Kinondo Secondary | 32 | 1 | 3.1 |
| Golini Secondary | 26 | 0 | 0 |
| Kichakasimba Secondary | 26 | 0 | 0 |
| Mvideni Secondary | 31 | 0 | 0 |
| Shimba Hills Seconadry | 50 | 0 | 0 |

**Table 3. Bivariate analyses of voluntary school-going adolescents' blood donors at KSBTC, January 2020-June 2021 (n = 613).**

| Variables | Number = 613 | Positive | Percent | Crude POR 95% CI | p value |
|---|---|---|---|---|---|
| **Age group (years)** | | | | | |
| 16–17 | 108 | 11 | 10.2 | 1.2 (0.60–2.43) | 0.59 |
| 18–25 | 502 | 43 | 8.6 | 1 | |
| **Sex** | | | | | |
| Female | 156 | 17 | 10.9 | 1.4 (0.76–2.54) | 0.29 |
| Male | 420 | 37 | 8.1 | 1 | |
| **Religion** | | | | | |
| Christianity | 275 | 25 | 9.1 | 1.1 (0.60–1.85) | 0.85 |
| Islam | 335 | 29 | 8.7 | 1 | |
| **Donation frequency** | | | | | |
| First timer | 513 | 48 | 9.4 | 1.5 (0.61–3.57) | 0.38 |
| Repeat | 92 | 6 | 6.5 | 1 | |

**POR, prevalence odds ratio, CI, confidence interval**.

On bivariate analyses, first-time blood donors were 1.5 times more likely to test positive for HBsAg compared to repeat donors (cPOR = 1.5, 95% CI 0.61–3.57, p-value 0.38). Females were 1.4 times more likely to test positive for HBsAg compared to male donors (cPOR = 1.4, 95% CI 0.76–2.54, p-value 0.29). There was a trend of 16–17 age group testing positive for HBsAg more frequently compared to 18–25 age group, though this difference was not statistically significant (cPOR = 1.2,95%CI = 0.60–2.45), p-value 0.59) (Table 3).

## Discussion

The main challenge with blood transfusion is blood safety besides blood shortage. However over the past years a reduction in transfusion transmitted HBV has been documented but there still exist the risk for viral infections transmitted post blood transfusion. This study confirms that HBV infection remains a safety concern in blood donated in this region. One possible explanation for this is that this region relies on blood supply from mobile voluntarily adolescents aged 16 to 24 years in learning institutions who are also at high risk due to their onset of active sexual and other practices that increase their exposure to the virus [2, 3]. In this study male donors were 89.1% while female donors were 10.9%. Higher proportion of male donors has also been reported in other studies including *Bartonjo et al* [15], who reported 72% were male donors, *Wamamba et al* reported 76% were male donors [16], and *Ali et al* reported a ratio of 1:5 were male voluntary young donors [17]. Sometimes voluntary donors among girls at this age could be deferred due to low hemoglobin levels as a result of heavy menstrual flows or even early pregnancies [18]. On the other hand we note that from the blood donation drives 60% of the secondary schools included were all boy schools which could also probably affect the ratio in our study.

The prevalence of HBV infections was approximately 9% which is higher than the prevalence reported among young adults (18–25 years) voluntary blood donors in the three Kenyan Counties of Siaya 4.4%, Homabay 3.4%, and Kisumu 2.6% [19]. However, the prevalence was lower than the study conducted in Nigeria among secondary students who had a prevalence of 18.4% [20]. These differences in zero-prevalence could be attributed to variations in geographical locations and the sensitivity of the screening techniques used in the detection of the HBsAg for the different studies. Other drivers of this difference may include the pre-donation screening tools employed by KNBTS, where prospective donors are given a risk assessment

questionnaire that asks about the past medical history of the potential donors and their sexual behavior which helps minimize the chances of receiving infected blood.

First-time blood donors were 50% more likely to test positive for HBsAg compared to repeat donors although this was not statistically significant. Several previous studies found varying results and reported a higher prevalence of HBV markers for the first-timer donors. A study in Sierra Leone reported 38.4-fold greater odds of having HBV in first-timer donors compared to repeat donors [21]. In Kenya, first-time donors were 1.7 times more likely to test positive for HBsAg [16]. This may be confounded by the fact that they are absolutely not aware of their status unlike the previous donors who could probably have prior knowledge from past screening. It is unlikely that if one donates blood once and tests positive for HBV that they will volunteer to donate again. The percentage of first-time volunteer blood being discarded was 17% in Bobo-Dioulasso but 23.1% and 23.9% in Ouagadougou and Fada'Ngourma, respectively [22].

In our study, the age group 16–17 years had a 20% higher chance of testing positive for HBsAg compared to the age group 18–25 years. In contrast to this study, teenagers aged (16–19 years)were 47% less likely to test positive for HBsAg compared to students above 20 years in three counties for Western Kenya [23]. The reported high prevalence among this age category is of concern, for it is expected that all of the study participants may have received childhood vaccination for HBV infections as Kenya introduced the vaccine in the year 2002 [24], thus, the data from this study might be used to evaluate the efficacy of HBV vaccine in the study area.

Female voluntary donors were 40% more likely to test positive for HBsAg compared to male donors. This is most likely because this study couldn't recruit enough females although similar studies in Ghana have reported a high prevalence of HBV infections among the female gender [25, 26]. In contrast, several other studies reported higher proportions of HBsAg positivity among male blood donors [16, 19, 27, 28]. The likely reasons for the different results were the different designs and settings employed by the different studies.

This study provides insights into the characteristics of potential risky blood donor populations, which is an important tool for blood safety and can be useful for policy formulation. This study employed a census-type sampling technique which minimized potential selection bias in sampling a well-defined study population and this should increase comparability. This study also had several limitations. The study did not conduct a follow-up on the individuals to monitor the disease progression of those with occult infections. Also, the study included only healthy voluntary blood donors who were attending secondary level of education; hence, our results may not reflect the overall prevalence of HBV infections in the general population of Kwale County. This might potentially underestimate infections; nevertheless, this study can provide proxy prevalence and predictors in this region when community-based surveys are not feasible. Another study limitation may be an intrinsic weakness in the diagnostic tests used. The use of serological techniques as opposed to nucleic acid-based techniques. However, the internal quality controls were performed for both positive and negative cases.

## Conclusion

This study reveals that the majority of the voluntary young blood donors were male, is blood group O positive, and that occult HBV infection remains a considerable safety concern in the age group of 16–25 years, first-time and female donors. Therefore, greater coverage of primary HBV vaccination should be encouraged in this area, and for those who have received their primary vaccine, a booster HBV vaccine could be provided in order to more effectively prevent HBV infection. This study recommends the KSBTC personnel's should adopt strategies to

increase repeat donors and encourage healthy first-timer donors to be regular donors should be intensified as such medical selection of blood donors may reduce the frequency of hepatitis B infections in blood donors, which will go a long way improving blood safety and availability. Also, these findings underline the need for confirmatory strategies to avoid blood wastage and to re-evaluate Hepatitis B infections prevalence in the study area among blood donors that may be overestimated.

## Supporting information

**S1 Dataset.**
(XLS)

## Acknowledgments

We would like to thank the Technical University of Mombasa, Kenya, for their support during the study. Special thanks go to the Kwale County Department of Health authorities for their collaboration. We also thank the health workers who conducted the blood drives and thereafter screened for transmissible transfusions infections, including John Kaluku, Raphael Kaplich, Mwanajuma Mbale, and Mushee Mohammed for their dedication and meticulous work. The authors are indebted with the following reviewers who took time to add their experience and knowledge on editing and reviewing the manuscripts: Katie White, Michael Bruce, Elizabeth Oele, Keli Gerken, and Erick Orimbo. The authors are indebted to African Field Epidemiology Network (AFENET) through Field Epidemiology and Laboratory Program, Kenya for support in paying publication fees. We extend a special thank you to all young voluntary blood donors for their altruism and heroic act of giving blood to those who need it.

The authors are indebted to African Field Epidemiology Network (AFENET) through Field Epidemiology and Laboratory Program, Kenya for support in paying publication fees.

## Author Contributions

**Conceptualization:** Peter Kitemi Wahome, Polly Kiende, Rocky Jumapili Nakazea, Narcis Mwakidedela Mwasowa, Gibson Waweru Nyamu.

**Data curation:** Polly Kiende, Narcis Mwakidedela Mwasowa, Gibson Waweru Nyamu.

**Formal analysis:** Polly Kiende, Gibson Waweru Nyamu.

**Funding acquisition:** Peter Kitemi Wahome, Polly Kiende.

**Investigation:** Peter Kitemi Wahome, Rocky Jumapili Nakazea, Narcis Mwakidedela Mwasowa, Gibson Waweru Nyamu.

**Methodology:** Peter Kitemi Wahome, Polly Kiende, Narcis Mwakidedela Mwasowa, Gibson Waweru Nyamu.

**Project administration:** Polly Kiende, Rocky Jumapili Nakazea.

**Supervision:** Polly Kiende, Narcis Mwakidedela Mwasowa, Gibson Waweru Nyamu.

**Validation:** Peter Kitemi Wahome, Polly Kiende, Rocky Jumapili Nakazea, Narcis Mwakidedela Mwasowa, Gibson Waweru Nyamu.

**Visualization:** Polly Kiende, Rocky Jumapili Nakazea, Narcis Mwakidedela Mwasowa, Gibson Waweru Nyamu.

**Writing – original draft:** Polly Kiende, Gibson Waweru Nyamu.

**Writing – review & editing:** Peter Kitemi Wahome, Polly Kiende, Rocky Jumapili Nakazea, Narcis Mwakidedela Mwasowa, Gibson Waweru Nyamu.

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
